# Osteosarcoma Arising as a Secondary Malignancy following Treatment for Hematologic Cancer: A Report of 33 Affected Patients from the Cooperative Osteosarcoma Study Group (COSS)

**DOI:** 10.3390/cancers16101836

**Published:** 2024-05-11

**Authors:** Stefan S. Bielack, Vanessa Mettmann, Daniel Baumhoer, Claudia Blattmann, Birgit Burkhardt, Christoph K. W. Deinzer, Leo Kager, Matthias Kevric, Christine Mauz-Körholz, Peter Müller-Abt, Dirk Reinhardt, Alexandru-Anton Sabo, Martin Schrappe, Benjamin Sorg, Reinhard Windhager, Stefanie Hecker-Nolting

**Affiliations:** 1Klinikum Stuttgart—Olgahospital, Stuttgart Cancer Center, Zentrum für Kinder-, Jugend- und Frauenmedizin, Pädiatrie 5 (Onkologie, Hämatologie, Immunologie), 70174 Stuttgart, Germany; vanessa.mettmann@umm.de (V.M.); c.blattmann@klinikum-stuttgart.de (C.B.); m.kevric@klinikum-stuttgart.de (M.K.); a.sabo@klinikum-stuttgart.de (A.-A.S.); b.sorg@klinikum-stuttgart.de (B.S.); s.hecker-nolting@klinikum-stuttgart.de (S.H.-N.); 2Klinik für Kinder- und Jugendmedizin, Pädiatrische Hämatologie und Onkologie, Universitätsklinikum Münster, 48149 Münster, Germany; birgit.burkhardt@ukmuenster.de; 3Knochentumor Referenzzentrum, Institut für Medizinische Genetik und Pathologie, Universitätsspital und Universität Basel, 4031 Basel, Switzerland; daniel.baumhoer@usb.ch; 4Abteilung Innere Medizin VIII—Medizinische Onkologie und Pneumologie, Universitätsklinikum Tübingen, 72076 Tübingen, Germany; christoph.deinzer@med.uni-tuebingen.de; 5St. Anna Kinderspital, Universitätsklinik für Kinder- und Jugendheilkunde der Medizinischen Universität Wien, 1090 Vienna, Austria; 6St. Anna Children’s Cancer Research Institute (CCRI), 1090 Vienna, Austria; 7Pädiatrische Hämatologie und Onkologie, Zentrum für Kinderheilkunde der Justus-Liebig-Universität, 35390 Gießen, Germany; christine.mauz-koerholz@paediat.med.uni-giessen.de; 8Medizinische Fakultät der Martin-Luther-Universität Halle-Wittenberg, 06120 Halle, Germany; 9Radiologisches Institut (Kinderradiologie), Zentrum für Kinder-, Jugend- und Frauenmedizin, Klinikum Stuttgart—Olgahospital, 70174 Stuttgart, Germany; p.mueller-abt@klinikum-stuttgart.de; 10Klinik für Kinderheilkunde III, Zentrum für Kinder- und Jugendmedizin, Universitätsmedizin Essen, 45147 Essen, Germany; dirk.reinhardt@uk-essen.de; 11Klinik für Kinder- und Jugendmedizin I, Campus Kiel, Universitätsklinikum Schleswig-Holstein, 24105 Kiel, Germany; m.schrappe@pediatrics.uni-kiel.de; 12Universitätsklinik für Orthopädie und Unfallchirurgie, Klinische Abteilung für Orthopädie, Medizinische Universität Wien, 1090 Vienna, Austria; reinhard.windhager@akhwien.at

**Keywords:** leukemia, lymphoma, osteosarcoma, secondary malignancy, chemotherapy, radiation therapy, prognosis

## Abstract

**Simple Summary:**

Osteosarcoma may arise as a secondary malignant neoplasm following a previous hematologic malignancy. We set out to identify potential risk factors, treatments given, and outcomes. Thirty-three eligible patients were identified in the database of our Cooperative Osteosarcoma Study Group. On average, the osteosarcomas developed close to a decade after the hematologic cancer, when patients were mostly still pediatric. Radiotherapy administered to battle the hematologic cancer seemed to be a major risk factor, and genetic tumor predispositions were identified in a subset of patients. Modern interdisciplinary osteosarcoma treatment, with adaptations for previous therapies, seemed feasible. The prognosis of the affected patients was inferior to patients without preceding cancers, but was far from being universally fatal. Additional malignancies complicated the course in a subset of patients. These results will help us to interpret the findings and to guide treatment in future patients.

**Abstract:**

Purpose: Osteosarcoma may arise as a secondary cancer following leukemias or lymphomas. We intended to increase the knowledge about such rare events. Patients and methods: We searched the Cooperative Osteosarcoma Study Group’s database for individuals who developed their osteosarcoma following a previous hematological malignancy. The presentation and treatment of both malignancies was investigated, and additional neoplasms were noted. Outcomes after osteosarcoma were analyzed and potential prognostic factors were searched for. Results: A total of 33 eligible patients were identified (male: 23, female: 10; median age: 12.9 years at diagnosis of hematological cancer; 20 lymphomas, 13 leukemias). A cancer predisposition syndrome was evident in one patient only. The hematological cancers had been treated by radiotherapy in 28 (1 unknown) and chemotherapy in 26 cases, including bone-marrow transplantation in 9. The secondary bone sarcomas (high-grade central 27, periosteal 2, extra-osseous 2, undifferentiated pleomorphic sarcoma of bone 2) arose after a median lag-time of 9.4 years, when patients were a median of 19.1 years old. Tumors were considered radiation-related in 26 cases (1 unknown). Osteosarcoma-sites were in the extremities (19), trunk (12), or head and neck (2). Metastases at diagnosis affected eight patients. Information on osteosarcoma therapy was available for 31 cases. All of these received chemotherapy. Local therapy involved surgery in 27 patients, with a good response reported for 9/18 eligible patients. Local radiotherapy was given to three patients. The median follow-up was 3.9 (0.3–12.0) years after bone tumor diagnosis. During this period, 21 patients had developed events as defined, and 15 had died, resulting in 5-year event-free and overall survival rates of 40% (standard error: 9%) and 56% (10%), respectively. There were multiple instances of additional neoplasms. Several factors were found to be of prognostic value (*p* < 0.05) for event-free (osteosarcoma site in the extremities) or overall (achievement of a surgical osteosarcoma-remission, receiving chemotherapy for the hematologic malignancy) survival. Conclusions: We were able to prove radiation therapy for hematological malignancies to be the predominant risk factor for later osteosarcomas. A resulting overrepresentation of axial and a tendency towards additional neoplasms affects prognosis. Still, selected patients may become long-term survivors with appropriate therapies, which is an argument against therapeutic negligence.

## 1. Introduction

The use of modern, often multi-modal, treatment designs may lead to long-term survival in a significant number of patients with both acute and chronic forms of leukemia or with lymphoma [1,2,3,4,5]. While this is a well justified cause of celebration, the intensive therapies required for such a cure do not come without costs [6,7,8]. Among the late side effects of otherwise successful cancer therapy, the development of secondary malignancies features as one of the most serious [9]. Among these, osteosarcoma, the most frequent primary malignancy of the bone [10], is one of the more common ones [11,12].

In addition to therapies used against hematological malignancy, an individual’s predisposition to develop multiple malignancies may also play a role. Li–Fraumeni syndrome acts as one example for such genetically encrypted cancer predisposition syndromes [13]. Recent studies have suggested genetic causes to underlie over 10–20% of osteosarcomas [14]. While the prognosis of patients with secondary osteosarcomas is generally believed to be poor, there may be relevant subgroups of patients who become long-term survivors with adequate therapies [11,12].

So far, there has been no dedicated analysis of osteosarcomas occurring secondarily to hematologic malignancies. Hence, neither the predisposing factors nor the disease outcome are very well defined. The large Cooperative Osteosarcoma Study Group’s (COSS) database [15] allowed us to extract such patients’ data and to perform detailed analyses of the presenting factors, the therapies used against both hematologic malignancies and secondary bone sarcoma, and the final outcomes.

## 2. Patients and Methods

### 2.1. Patient Selection and Data Collection

The database of the Cooperative Osteosarcoma Study group COSS as of 23 May 2023 was screened for all patients who developed bone sarcoma following a previous diagnosis of a leukemia or lymphoma.

Details of the group’s recruitment strategies and treatment protocols have been described previously [15,16,17,18]. By necessity, all information regarding the hematological malignancies was collected retrospectively. Data on patient demographics as well as bone sarcoma characteristics, bone sarcoma therapy, and follow-up information were collected and coded prospectively as described [16]. Further information was derived from status report forms; radiology, pathology, and surgery reports; as well as progress letters, if available at the study center.

All COSS study and registry protocols were performed in accordance with the Code of Ethics of the World Medical Association (Declaration of Helsinki) and approved by the appropriate ethics committee (please see Institutional Review Board Statement).

The diagnosis of hematologic malignancy was made according to the standards of the treating institution, and may have been made according to multi-institutional guidelines or guidance. For the diagnosis of bone sarcoma, histologic investigations and suitable immunohistochemistry techniques were used according to local practice and, if material was available, by a COSS-referenced pathologist. The staging procedures prescribed by all COSS protocols included conventional radiography of the tumor, chest radiography, computed tomography of the thorax, and a whole-body ^99^Tc-methylene-diphosphonate bone scan. Computed tomography, magnetic resonance tomography of the primary site, and positron emission tomography were used according to time and availability. The treatment followed the guidelines and guidance detailed in the COSS protocol active at the time of enrolment. In brief, all patients were to receive neo-adjuvant and adjuvant chemotherapy. The drugs used varied with time, but almost always included high-dose methotrexate, doxorubicin, and cisplatin. Ifosfamide was prescribed for a relevant minority of patients. Fewer individuals received other drugs. Adjustments for previous therapies were allowed.

Local treatment of osteosarcoma was surgical whenever feasible. Wide margins according to Enneking [19] were recommended whenever feasible. The responses of the primary tumors to preoperative chemotherapy were graded according to Salzer-Kuntschik et al., a good response being defined as <10% viable tumor cells in the resected specimen [20]. A patient was considered to have achieved complete surgical remission if all tumor foci were removed macroscopically.

Follow-up guidelines for osteosarcoma varied according to the time of recruitment and the protocol. In general, the search for local recurrence was performed by means of conventional X-ray at least every 3 months for 4–5 years after treatment, and only in case of clinical suspicion thereafter. Lung metastases were to be searched for by conventional chest X-rays, recommended every 4–8 weeks during years 1–2, every 8–12 weeks during years 3 and 4, and every 6 months from year 5 to years 8–10. Later follow-up was recommended, but left to the treating physician’s discretion.

### 2.2. Statistical Analyses

All patients were evaluated on an intention-to-treat basis. The starting date for all survival analyses was the date of osteosarcoma diagnosis. Follow-up was until the time of death, with the causes being noted. Overall survival was calculated until death or last patient information, whichever was appropriate. Event-free survival was calculated until first recurrence, date last known to be alive, or death, whichever occurred first. Additional malignancies, regardless of where they occurred, were noted, but not counted as events.

Survival estimates were calculated according to Kaplan and Meier [21], with 95% confidence estimates. A comparison of survival expectancies between unrelated cohorts was made using the log-rank test [22]. *p*-values ≤ 0.05 were considered significant, and no correction for multiple testing was made. Statistical analyses were carried out using the SPSS statistical software packet (IBM Corp. Released 2022. IBM SPSS Statistics for Windows, SPSS version 29.0.0.0., IBM Corp.: Armonk, NY, USA).

## 3. Results

### Patients

The search revealed 34 patients who only developed osteosarcoma following a previous hematological malignancy. One of these was excluded, as registration only occurred upon osteosarcoma relapse (see CONSORT-diagram, Figure 1). This left 33 patients from 27 institutions which registered either 1 (n = 22), 2 (n = 4), or 3 (n = 1) patients to form the basis for all further analyses (Table 1). The cohort consisted of 23 (70%) males and 10 (30%) females.

A cancer predisposition syndrome, Rothmund–Thomson syndrome, was genetically proven in 1/33 (3%) patients. There were no cases of cancer prior to the hematological malignancy. A benign tumor, eosinophilic granuloma, was documented in 1/33 (3%).

The hematologic malignancy arose when patients were a median of 12.6 (range: 0.8–58.2) years old. These were:-20 (61%) lymphomas (13 Hodgkin (HL), 7 non-Hodgkin (NHL), 3 diffuse large B-cell (DLBCL), 1 anaplastic large-cell (ALCL), 1 lymphoblastic, 1 mucosa-associated lymphoid tissue (MALT), and 1 unspecified NHL). Among the 20 lymphomas, 16 (80%) were known to have involved the trunk, 4 (20%) the head and neck (incl. 1 with involvement of both regions), and 1 (5%) an extremity.-13 (29%) leukemias (12 acute lymphoblastic (ALL)—7 B-precursor, 4 T-, 1 unspecified; 1 acute myeloid leukemia (AML)).

Radiotherapy was used to treat 28/32 (88%) hematologic malignancies (1 unknown). The maximum radiotherapy dosage was known for 22/28 (79%) patients and was 32 (12–66) Gy, including total-body irradiation in 9 individuals (all with ALL). In addition, 26/33 (79%) patients had received chemotherapy, while 7/33 (21%; 5 HL, 1 DLBCL, 1 MALT-lymphoma) had not. A total of 6/26 (23%) chemotherapeutically treated patients were known to have received additional chemotherapy for recurrence (5 ALL, 1 NHL). An allogeneic bone marrow transplantation was known to have been performed in 9/33 (27%; 8 ALL, 1 AML) individuals.

In the time period between leukemia or lymphoma and osteosarcoma, 7/33 (21%) patients had suffered from a distinct neoplasm. These were:-1 benign tumor (adenoma of the thyroid treated by surgery);-3 borderline tumors (2 basal cell carcinomas of the skin, 1 phylloides tumors of the breast, all treated by surgery only);-1 solid malignancy (1 breast cancer treated by surgery and local radiotherapy);-2 hematologic malignancies (1 untreated large-cell NHL, 1 chronic lymphocytic leukemia treated by chemotherapy).

The secondary osteosarcomas (Table 2) arose after a median lag time of 9.4 (0.5–41.6) years from the patient’s first hematologic malignancy. The affected patients were then a median of 19.1 (6.8–69.6) years old. A total of 2/33 (6%) were still in their first and 17/33 (52%) in their second decade of life, while 14 (42%) were older. Osteosarcoma histologies were reported as high-grade central in 27/33 (82%) and periosteal, extra-osseous, and undifferentiated pleomorphic sarcoma (UPS) of bone in in 2/33 (6%) of patients, each. Reference pathology confirming this diagnosis was available for 25/33 (76%) cases.

Among the 32/33 (97%) patients with relevant information, the bone tumors were considered to have been radiation-related in 26/32 patients (81%; 24 in-field, 2 immediately adjacent).

Information on the duration of symptoms (pain and/or swelling) was available for 23/33 (70%) patients. In these, it lasted a median of 27 (6–272) days.

Sites affected by osteosarcoma were the extremities in 19/33 (58%; femur: 7 (proximal: 1, distal: 6), tibia: 5 (all proximal), humerus: 5 (proximal: 4, distal: 1), distal (radius: 1, foot: 1); the trunk in 12/33 (36%; clavicle: 5, rib: 2, scapula: 1, thoracic spine: 1, pelvis (ilium): 2, groin extraosseus: 1); and the head and neck in 2/33 (6%; mandibular: 1, neck extraosseous: 1) patients.

Metastases at diagnosis were reported for 8/33 (24%) patients. These affected the lungs only in five, the bones only in one, and both combined in two cases.

Information on local and systemic osteosarcoma treatment was available for 31/33 (94%) patients. Of these, all (100%) were treated systemically (5 preoperatively only, 5 postoperatively only, and 21 combined). The median duration of reported systemic therapy was 236 (range: 20–605) days. Drugs known to have been administered were cisplatin: 29/31 (94%), doxorubicin: 28/31 (90%, 1 of these PEGylated), high-dose methotrexate: 26/31 (84%), ifosfamide: 23/31 (74%), etoposide: 10/31 (32%), carboplatin: 8/31 (26%), and others: 4/31 (13%). Additionally, 2/31 (6%) individuals were subjected to targeted therapy, and 1/31 (3%) to PEGylated interferon.

Local therapy was administered via surgery in 27/31 (87%) patients. Among the 19 extremity primaries, 18 (95%) were operated upon, by limb-salvage in 14/18 (78%) cases. Radiotherapy was administered to 3/30 (10%; 2 in addition to, 1 instead of surgery) secondary osteosarcomas. For those 23/27 (85%; 4 adjuvant therapy only) patients who were operated upon with documented administration of preoperative chemotherapy, the histologic response was known for 18/23 (78%). It was reported as good or poor in 9/18 (50%) patients.

Follow-up for events lasted a median of 2.3 (1 day–12.0) years from osteosarcoma. During this period, events, as defined, occurred in 21/33 (64%) patients after a median of 1.6 (1 day–7.2) years. The events were as follows: failure to achieve surgical remission: 8, metastases only: 10 (pulmonary: 9, mediastinal: 1), local recurrence only: 1, combined local recurrence and pulmonary metastases: 1, death of further malignancy: 1. In total, 5/33 (15%) patients suffered further tumors or hematologic cancers after the osteosarcoma. These were not counted as events by definition. They included T-ALL, B-NHL, MDS, esophageal carcinoma, and basal cell carcinoma of the skin (1 each).

The median follow-up for survival was 3.9 (0.3–12.0) years after bone tumor diagnosis for all 33 patients and 5.0 (1 day–12.0) years for 18/33 (55%) survivors. Among these 18 survivors, 4 were alive without having achieved surgical remission, and the remaining 14 were in 1st (12), 2nd (1), or 3rd (1) surgical remission. A total of 15/33 (45%) individuals had died after a median of 3.1 (0.4–7.4) years since osteosarcoma. Causes of death were: failure to achieve osteosarcoma remission (2), osteosarcoma recurrence (10; 6: 1st, 4: 2nd), secondary malignancy (2; T-ALL: 1, MDS: 1), and unknown without having achieved osteosarcoma remission (1).

Event-free survival, as defined, was found to be 73% (standard error: 8%), 63% (9%), and 40% (9%) at 1, 2, and 5 years after osteosarcoma diagnosis, respectively. The corresponding values for overall survival were 90% (5%), 87% (6%), and 56% (10%) (Figure 2). Potential predictive and prognostic factors are presented in Table 3: Patients who were able to achieve surgical osteosarcoma remission fared better than those who were not (*p* = 0.017 for overall survival, calculation for event-free survival not possible by definition). Patients who had received chemotherapy for their hematologic malignancy fared better than those that had not (*p* = 0.024). For event-free survival, only an osteosarcoma site in the extremities was found to have a favorable result (*p* = 0.011). An absence of primary osteosarcoma metastases was of borderline significance for both overall (*p* = 0.050) and event-free (*p* = 0.063) survival.

## 4. Discussion

This report describes secondary osteosarcomas in individuals previously treated for hematologic cancers. It proves that radiation administered against the former is undisputedly a major predisposing factor for the latter. It also demonstrates which patients with this unfortunate combination of malignancies may still be salvaged, and how this may be accomplished.

To our knowledge, this series of 33 affected individuals represents the largest specific series so far. For this report, we chose to include tumors classified as undifferentiated pleomorphic sarcomas of the bone in the analyzed cohort because of their striking biological similarity to osteosarcoma [23].

It is notable that only one of our doubly affected patients was reported to have carried a genetically proven cancer predisposition, in this case Rothmund–Thomson syndrome. We cannot completely ascertain that testing was adequately performed in all remaining cases. Also, we might not have been supplied with all relevant information if it was. However, our data may still argue against a major role of genetic predispositions in this context. It is also notable that none of our patients had contracted any cancer prior to having experienced their hematologic malignancy. Both facts seem to argue against a frequent genetic basis for the development of these multiple cancers.

There was a striking predominance of males in this series. Osteosarcoma is generally known for its male predominance [10]. This may be somewhat exaggerated by the epidemiology of some hematological malignancies. Taken together, this sufficiently explains the observed distribution of genders.

The age at which the leukemias and lymphomas arose seems quite typical for the diseases in general. With ours being a predominantly pediatric group, we noted a median of less than 13 years of age in our series. However, some of the patients, in particular those with certain types of lymphoma, were considerably older.

Among the lymphomas which were predecessors to later osteosarcomas, Hodgkin lymphoma predominated over other subtypes. It may be of particular interest that almost all of the bone tumors in former lymphoma patients arose within or adjacent to formerly irradiated fields. Total-body irradiation, administered as a part of bone marrow transplantation, seemed to have played an unwholesome role in the further course of our leukemia patients. Both forms of radiotherapy were probably very well indicated in some lymphoma or high-risk leukemia patients, or to treat leukemia relapses. Our analysis demonstrates, however, that employing this form of treatment does not come without its costs.

Among the leukemias, we observed a striking predominance of the acute lymphoblastic over the acute myeloblastic type. Both varying incidences as well as the much poorer long-term survival probability associated with the latter contributed to this finding, and differing strategies of radiation use for CNS-prophylaxis may have as well.

It is of relevance that the lag time between the original hematologic malignancy and the later bone sarcoma lasted several decades in some patients, and was more than ten years in almost half of the cases. This impressively demonstrates why follow-up periods should not end prematurely: Long-term care is essential.

Screening for the development of osteosarcomas may be especially relevant if the therapy administered against hematologic malignancies includes any form of radiotherapy, and if total-body irradiation is involved. After all, the vast majority of patients in the reported series had received such therapy.

The secondary osteosarcomas we observed differed from primary tumors of the same kind in several aspects: Primary osteosarcomas most frequently arose in the extremities of adolescents [10]. Here, the median age of patients affected by secondary osteosarcomas was found to be almost 20 years. We also observed a considerable number of axial and even some head and neck primaries. This must be attributed to the prior use of radiotherapy to those areas. The proportion of patients with primary osteosarcoma metastases was certainly not lower than expected.

Some deviations from standard osteosarcoma drugs and protocols were probably unavoidable given previous exposures, particularly to anthracyclines [24]. Most patients were, however, able to once again withstand what seemed to be adequate systemic therapy. This included preoperative treatment in a majority of individuals. The resulting response rate in the evaluable osteosarcomas seemed somewhat representative of the disease as a whole, and was not different from what might be expected for primary tumors of this kind [10].

We [11] and others [12] have previously been able to demonstrate that osteosarcomas arising as secondary malignancies may be cured in a subset of patients. This paper extends the observation to hematological malignancies. Approximately half of all affected patients may survive for another five years, with most, but not all, of those being disease-free. This is of utmost importance for teams having to face such unfortunate situations, as it argues against therapeutic negligence and may act as a motivation. Even though this is a comparatively large series, the analysis of potential predictive or prognostic factors is still impeded by limited patient numbers. It seems that those factors most strongly influencing prognosis in the general osteosarcoma patient, like extremity site or the ability to achieve surgical remission, also emerge here, as there might an absence of primary osteosarcoma metastases. It seems to be that the decision to administer chemotherapy for hematological cancer was only a substitute for other prognostically positive factors. A tendency of other hematologic diseases to appear in extremity sites compared to Hodgkin lymphomas, which were irradiated, may be one of these factors. As a cautionary note, hematologic cancers and later osteosarcomas were not all that was experienced by the patients. As demonstrated by the unrelated tumors and hematologic malignancies arising both before and after the osteosarcoma, an increasing treatment burden does not come without additional costs. This must be considered when interpreting the observed survival rates.

In summary, we were able to analyze an unprecedented number of osteosarcomas which developed after prior treatment for a hematologic malignancy. We were also able to provide data suggesting that radiotherapy administered against the former is a major risk factor for the development of the latter. Importantly, systemic treatment of the bone sarcoma often proved feasible despite pretreatment. Patients with osteosarcomas arising after a previous leukemia or lymphoma should not be denied potentially curative therapy.

## 5. Conclusions

Osteosarcoma arising as a secondary cancer after leukemia or lymphoma is certainly bad news. However, it need not be universally fatal if treated appropriately. Careful consideration of variables such as potential cancer predisposition syndromes, previous cancer therapies, and drugs administered to particular existing radiation fieldsand to all areas involving the osteosarcoma is mandatory in order to assess each individual situation. With an individualized treatment regimen which takes into account all of these variables, it may be possible to treat even these unfortunate patients with curative intent. Long-term survival also necessitates long-term care, as health problems such as, but not limited to, further cancers may arise during observation. The good news is that such long-term survival is a real and distinct possibility!

## Figures and Tables

**Figure 1 cancers-16-01836-f001:**
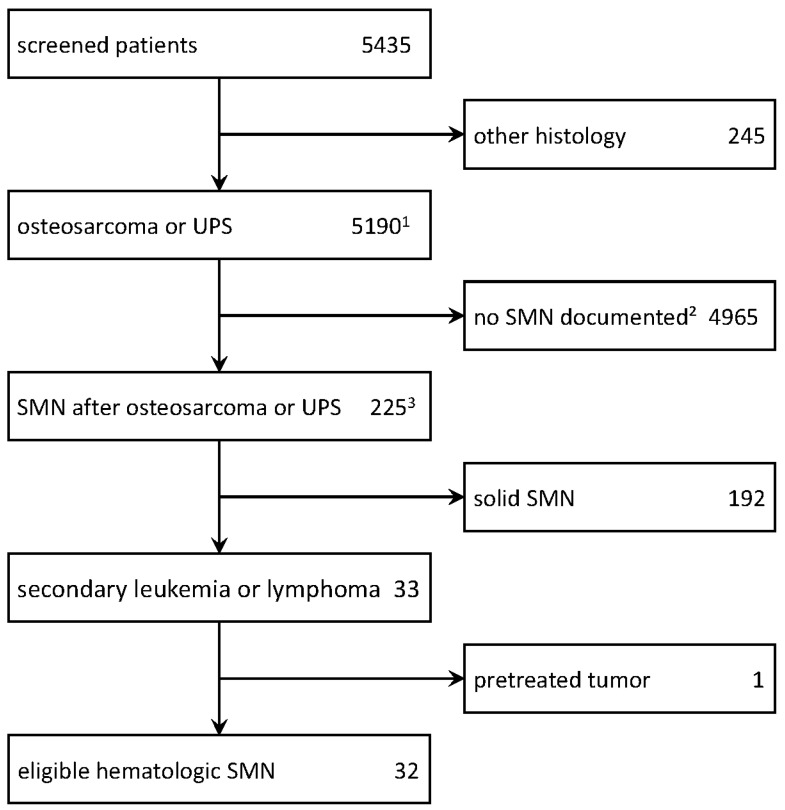
CONSORT diagram. Abbreviations: UPS = undifferentiated pleomorphic sarcoma of bone; SMN = secondary malignant disease; 1—5037 osteosarcomas, 153 UPS; 2—after diagnosis of osteosarcoma or UPS; 3—non-melanoma skin cancers excluded.

**Figure 2 cancers-16-01836-f002:**
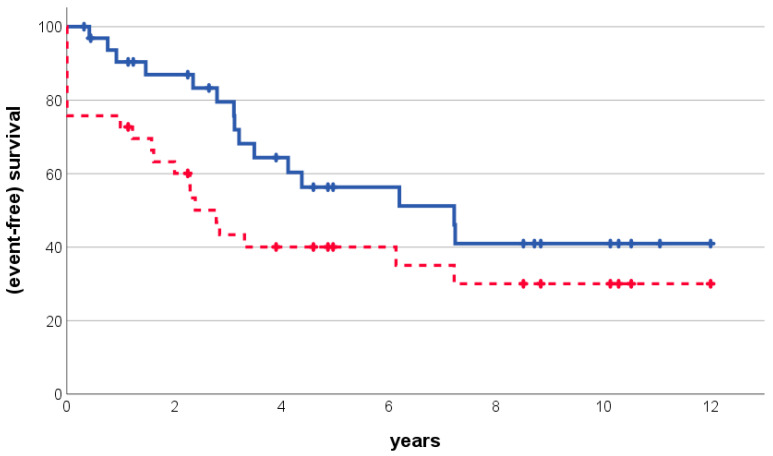
Overall (solid line) and event-free (dashed line) estimates of survival after the diagnosis of secondary osteosarcoma following hematological malignancy.

**Table 1 cancers-16-01836-t001:** Patients, hematological malignancies, and additional tumors preceding osteosarcoma. Ages are given in years at diagnosis of first hematologic malignancy.

No.	Gender	Hematological Malignancy				Intercurrent
		Age	Type	Radiotherapy	Chemotherapy	Neoplasm
1	male	14.2	HL	yes	none	none
2	male	2.9	NHL (NFS)	yes	yes	none
3	male	29.7	HL	yes	none	none
4	male	12.6	HL	yes	yes	none
5	male	16.6	HL	yes	yes	none
6	male	10.8	ALL (B-precursor)	yes (TBI)	yes	none
7	female	20.8	HL	yes	none	none
8	male	15.1	HL	yes	yes	none
9	male	5.6	ALL (B-precursor)	none	yes	none
10	male	7.4	ALL (NFS)	yes (TBI)	yes	none
11 ^1^	male	9.4	NHL (ALCL)	none	yes	NHL (B-large-cell)
12	male	5.8	HL	yes	yes	none
13	female	10.5	ALL (T-)	yes (TBI)	yes	none
14	male	12.9	ALL (T-)	yes	yes	none
15	male	33.4	HL	yes	none	basal-cell carcinoma
16	male	22.4	HL	yes	yes	none
17	female	49.8	NHL (DLBCL)	yes	yes	none
18	male	37.1	HL	yes	none	none
19	male	0.8	ALL (B-precursor)	yes (TBI)	yes	none
20	female	22.6	HL	yes	yes	none
21	male	58.2	NHL (MALT)	yes	none	CLL
22	female	38.1	NHL (DLBCL)	yes	yes	none
23	male	3.3	ALL (T-)	yes (TBI)	yes	none
24	male	15.0	HL	none	yes	none
25	male	1.4	ALL (B-precursor)	yes (TBI)	yes	thyroid adenoma
26	female	1.2	ALL (B-precursor)	yes (TBI)	yes	none
27	male	9.4	ALL (T-)	yes (TBI)	yes	none
28	female	45.7	NHL (DLBC)	yes	none	breast cancer
29	male	11.1	NHL (lymphoblastic)	yes	yes	basal-cell carcinoma
30 ^2^	female	5.4	ALL (B-precursor)	none	yes	none
31 ^2^	female	12.4	ALL (B-precursor)	yes (TBI)	yes	phylloides tumor
32	female	4.0	AML	ND	yes	none
33	m	15.3	HL	yes	yes	none

Abbreviations: Lymphomas: HL = Hodgkin lymphoma, NHL = non- Hodgkin lymphoma (ALCL = anaplastic large-cell lymphoma, DLBC = diffuse large B-cell lymphoma, MALT = mucosa-associated lymphoid tissue lymphoma). Leukemias: ALL = acute lymphoblastic leukemia, AML = acute myeloblastic leukemia, CLL = chronic lymphoblastic leukemia no. = patient number, TBI = total body irradiation, NFS = not further specified. 1—proven case of Rothmund–Thomsen syndrome. 2—clinical suspicion of Li–Fraumeni syndrome (genetics not provided).

**Table 2 cancers-16-01836-t002:** Characteristics of the 33 secondary leukemias and patient outcomes. Response of the osteosarcoma to preoperative chemotherapy, graded according to Salzer-Kuntschik et al. [20]. 1—without ever achieving a surgical remission of osteosarcoma; 2—in 1st surgical remission of osteosarcoma; 3—in remission of the later secondary malignancy; 4—marginal zone lymphoma.

No.	Inter-	Osteosarcoma								Follow-up		
	val	Age	Type	RX-rel	Prim mets	Surg	Response	Rx	Chem	Years	Further neopl	Outcome
1	5.5	19.7	HGC	yes	none	yes	good	none	yes	4.1	none	DOD 2nd rec
2	9.8	12.7	EOS	yes	none	no	NOP	none	yes	0.8	none	DOD no CR
3	18.0	47.7	EOS	yes	none	yes	POP	none	yes	3.1	none	DOD 1st rec
4	4.6	17.2	HGC	yes	none	yes	good	none	yes	4.6	none	alive CR1
5	19.0	35.6	HGC	yes	none	yes	NFS	none	yes	3.1	none	DOD 1st rec
6	8.2	19.0	HGC	yes	none	no	NOP	yes	yes	7.2	MDS	DOD SMD ^1^
7	17.7	38.5	HGC	yes	none	yes	poor	none	yes	1.5	none	DOD 1st rec
8	3.9	18.9	HGC	yes	none	yes	good	none	yes	11.0	none	alive CR2
9	7.5	13.1	HGC	no	lung	yes	good	none	yes	6.2	none	DOD 2nd rec
10	8.8	16.1	POS	yes	bone	yes	poor	none	yes	4.4	none	DOD 1st rec
11	5.3	14.7	HGC	no	none	yes	good	none	yes	7.2	T-ALL	DOC SMD ^2^
12	13.3	19.1	HGC	yes	none	yes	poor	none	yes	8.7	none	alive CR3
13	3.2	13.6	HGC	yes	lung	yes	poor	none	yes	2.4	none	DOD 1st rec
14	7.0	19.8	HGC	no	lung	yes	NFS	none	yes	3.5	none	DOD 2nd rec
15	27.1	60.5	HGC	yes	none	yes	poor	none	yes	12.0	SCC	alive CR1 ^3^
16	16.1	38.6	UPS	no	none	yes	good	none	yes	8.8	BCC	alive CR1 ^3^
17	6.9	56.7	UPS	yes	none	yes	POP	yes	yes	10.5	none	alive CR1
18	24.4	61.5	HGC	yes	lung, bone	yes	POP	none	yes	0.9	none	DOD no CR
19	13.7	14.5	HGC	yes	none	yes	good	none	yes	10.1	none	alive CR1
20	7.7	30.3	HGC	yes	none	no	NFS	NFS	NFS	0.4	none	alive no CR
21	11.4	69.6	HGC	yes	lung, bone	no	NFS	NFS	NFS	0.4	B-NHL ^4^	DUC no CR
22	11.2	49.3	HGC	yes	none	yes	POP	none	yes	1.1	none	alive CR1
23	13.9	17.2	HGC	yes	none	yes	poor	none	yes	2.8	none	DOD 2nd rec
24	0.5	15.5	HGC	no	lung	yes	good	none	yes	10.3	none	alive CR1
25	16.6	18.1	POS	yes	none	yes	poor	none	yes	3.9	none	alive CR1
26	5.6	6.8	HGC	yes	none	yes	poor	none	yes	8.5	none	alive CR1
27	9.5	18.9	HGC	yes	none	yes	NFS	none	yes	5.0	none	alive CR1
28	19.3	65.0	HGC	yes	none	yes	poor	none	yes	2.3	none	alive CR1
29	41.6	52.7	HGC	yes	lung	yes	NFS	yes	yes	2.6	none	alive no CR
30	9.3	14.8	HGC	no	none	yes	NFS	none	yes	4.9	none	alive CR1
31	9.4	21.9	HGC	yes	none	no	NOP	none	yes	1.2	none	alive no CR
32	6.0	9.9	HGC	NFS	none	yes	good	none	yes	3.2	none	DOD 1st rec
33	7.8	23.1	HGC	yes	none	no	NOP	none	yes	0.3	none	alive no CR

Header: No. = patient number; Interval and Age at osteosarcoma diagnosis, given in years; RX-rel = considered radiation-related; Prim mets = primary metastases at diagnosis of osteosarcoma; Surg = surgery; RX = radiation therapy; Chem = chemotherapy; Further neopl = further neoplasm after diagnosis of osteosarcoma. Osteosarcoma subtypes: HGC = high-grade central osteosarcoma, EOS = extra-osseous osteosarcoma, POS = periosteal osteosarcoma, UPS = undifferentiated pleomorphic sarcoma of bone. Response: NOP = no tumor operation, POP = primary tumor operated on before the start of chemotherapy. Further neoplasms: MDS = myelodysplastic syndrome, T-ALL = T-cell acute lymphoblastic leukemia, SCC = cutaneous squamous cell carcinoma, BCC = cutaneous basal cell carcinoma, B-NHL = B-cell non-Hodgkin lymphoma. Outcome: CR = complete remission, rec = recurrence, DOD = dead of osteosarcomatous disease, DOC = death of complications, DUC = death of unknown causes, SMD = secondary malignant disease arising after osteosarcoma diagnosis. NFS = not further specified.

**Table 3 cancers-16-01836-t003:** Survival expectancies according to clinical variables.

	Patients (%)	5-Year Overall Survival		5-Year Event-Free Survival	
		Estimate (SE)	*p* ^1^	Estimate (SE)	*p* ^1^
**all**					
all eligible patients	33 (100%)	56% (10%)	-	40% (9%)	-
**gender**					
male	23 (70%)	57% (11%)	0.831	39% (10%)	0.71
female	10 (30%)	51% (20%)		44% (18%)	
**age at leukemia/lymphoma**					
below 12 years	15 (45%)	64% (13%)	0.991	40% (13%)	0.983
12 years and above	18 (55%)	48% (14%)		40% (12%)	
**type of hematological malignancy**					
lymphoma	20 (61%)	57% (13%)	0.856	42% (12%)	0.735
leukemia	13 (39%)	57% (15%)		39% (14%)	
**chemotherapy for hematological malignancy**					
yes	26 (79%)	67% (10%)	0.024	45% (10%)	0.477
no	7 (21%)	19% (17%)		19% (17%)	
**interval hematological malignancy to osteosarcoma**					
below 10 years	19 (58%)	63% (12%)	0.736	42% (11%)	0.967
10 years or more	14 (42%)	49% (15%)		36% (14%)	
**age at osteosarcoma**					
below 18 years	12 (36%)	58% (14%)	0.797	50% (14%)	0.265
18 years and above	21 (64%)	55% (13%)		34% (11%)	
**osteosarcoma in radiation-field ^2,3^**					
yes	26 (81%)	51% (11%)	0.596	35% (10%)	0.273
no	6 (19%)	82% (15%)		67% (19%)	

**osteosarcoma site**					
extremity	19 (58%)	56% (11%)	0.225	56% (12%)	0.011
trunk or head and neck	14 (42%)	40% (17%)		18% (11%)	
**primary osteosarcoma metastases**					
absent	25 (76%)	65% (11%)	0.05	49% (10%)	0.063
present	8 (24%)	31% (18%)		13% (12%)	
**macroscopic surgical remission**					
achieved	25 (76%)	60% (10%)	0.017	NA	
not achieved	8 (24%)	51% (20%)			
**additional malignancies** ^4^					
no	28 (85%)	52% (11%)	0.807	38% (9%)	0.893
yes	5 (15%)	80% (18%)		60% (22%)	

1—log-rank test; 2—including total-body irradiation; 3—1 patient without information; 4—at any time, only fully malignant diseases considered. NA = not applicable.

## Data Availability

The data that support the findings of this study are available from the corresponding author upon reasonable request.

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
