# Peer review of "Osteosarcoma Arising as a Secondary Malignancy following Treatment for Hematologic Cancer: A Report of 33 Affected Patients from the Cooperative Osteosarcoma Study Group (COSS)"

_cancers, 2024, doi:10.3390/cancers16101836_

Round 1
Reviewer 1 Report
Comments and Suggestions for Authors
Title:
Osteosarcoma Arising as a Secondary Malignancy Following 2 Treatment for a Hematologic Cancer. A Report of 33 Affected 3 Patients from the Cooperative Osteosarcoma Study Group 4 (COSS)
Outlines:
The authors retrospectively review 33 patients from COSS database to evaluate the clinical features and outcome of osteosarcoma arising as a secondary cancer post treatment for previous hematological malignancy. They concluded that radiotherapy for hematological malignancies is a predominant risk factor for developing secondary osteosarcoma.
This is a nice article with a large number of patients specifically with the diagnosis of osteosarcoma arising as a secondary cancer post treatment for previous hematological malignancy. Here are some questions and suggestions for the authors.
1. The definition of event is confusing. In the methods session, event-free survival was calculated until first recurrence, date last known to be alive, or death, whichever occurred first. But in the result session, events were: failure to achieve a surgical remission 8, metastases only 10, and others. I am not sure if the 18 patients (failure to achieve a surgical remission and metastases only) had ever achieved remission or they had partial response/progressive disease after therapy. Probably, it is better to analyze as disease-free survival or progression-free survival.
2. Among 33 patients, 23 patients had been diagnosed with hematological malignancies in childhood (< 18 years of age). Twelve of 23 patients developed secondary osteosarcoma before age of 18, and 21 of 23 patients developed secondary osteosarcoma before age of 30 (young adulthood). I suggest the authors to have discussion on its impact on long-term follow-up of childhood cancer survivors.
3. Which patient was diagnosed with Rothmund-Thomson syndrome? Please indicate the case in the table or in the text.
4. Seven patients had other type of neoplasms between hematological malignancies and secondary osteosarcoma. What kind of treatment did they receive? Did the treatment for other type of neoplasms contribute to the development of osteosarcoma?
Author Response
We would like to thank all three reviewers for their efforts! We have tried to address their comments and suggestions as follows:
Reviewer 1.
Comment 1
The authors retrospectively review 33 patients from COSS database to evaluate the clinical features and outcome of osteosarcoma arising as a secondary cancer post treatment for previous hematological malignancy. They concluded that radiotherapy for hematological malignancies is a predominant risk factor for developing secondary osteosarcoma.
This is a nice article with a large number of patients specifically with the diagnosis of osteosarcoma arising as a secondary cancer post treatment for previous hematological malignancy. Here are some questions and suggestions for the authors.
Answer 1
We would like to thank the reviewer for the very positive reception of this analysis!
Comment 2
The definition of event is confusing. In the methods session, event-free survival was calculated until first recurrence, date last known to be alive, or death, whichever occurred first. But in the result session, events were: failure to achieve a surgical remission 8, metastases only 10, and others. I am not sure if the 18 patients (failure to achieve a surgical remission and metastases only) had ever achieved remission or they had partial response/progressive disease after therapy. Probably, it is better to analyze as disease-free survival or progression-free survival.
Answer 2
We agree with the reviewer that disease-free survival or progression-free survival might be suitable alternatives. For the latter, we simply do not collect the data for progression of existing lesions, which would be relevant for patients never achieving a remission. For event-free survival, we counted osteosarcoma-related events as well as death of any cause, simply because we, as an osteosarcoma-dedicated group, have the data to do so. This would not be the case for events related to other cancers, preceding or following osteosarcoma.
We have used one and only one particular calculation-method for all our analyses of event-free survival ever since analyzing prognostic factors for osteosarcoma in general (J Clin Oncol 2002, one of the 40 most frequently cited papers of that particular journal’s first 40 years of publication). We would ask to please be allowed to also use this method for the analysis presented here.
Comment 3
Among 33 patients, 23 patients had been diagnosed with hematological malignancies in childhood (< 18 years of age). Twelve of 23 patients developed secondary osteosarcoma before age of 18, and 21 of 23 patients developed secondary osteosarcoma before age of 30 (young adulthood). I suggest the authors to have discussion on its impact on long-term follow-up of childhood cancer survivors.
Answer 3
We believe that the manuscript - as originally submitted - already includes such a discussion. After all, it reads (paragraph 8 of the discussion):
“It is of relevance that the lag-time between the original hematologic malignancy and the later bone sarcoma extended several decades in some patients and was more than ten years in almost half of the cases. This impressively demonstrates why fol-low-up should not end prematurely: Long-term care is essential.”
Therefore, we have decided to not further amend the text and hope the reviewer can agree.
Comment 4
Which patient was diagnosed with Rothmund-Thomson syndrome? Please indicate the case in the table or in the text.
Answer 4
We regret that we neglected to include which patients were affected by hereditary tumor-predisposition syndromes! Notations explaining who carried the Rothmund-Thomsen syndrome as well as which two patients were suspected of being affected by the Li-Fraumeni syndrome were added to the revised table 1. We regret our omission!
Comment 5
Seven patients had other type of neoplasms between hematological malignancies and secondary osteosarcoma. What kind of treatment did they receive? Did the treatment for other type of neoplasms contribute to the development of osteosarcoma?
Answer 5
The treatment received for these malignancies was added to paragraph 5 of the Results section, which now reads:
“In the time period between leukemia or lymphoma and osteosarcoma, 7/33 (21%) had suffered from a distinct neoplasm. These were:
- 1 benign tumor (adenoma of the thyroid treated by surgery)
- 3 borderline tumors (2 basal-cell carcinomas of the skin, 1 phylloides tumor of the breast, all treated by surgery only)
- 1 solid malignancy (1 breast cancer treated by surgery and local radiotherapy)
- 2 hematologic malignancies (1 untreated large-cell NHL, 1 chronic lymphocytic leukemia treated by chemotherapy).”
We do not believe that the treatment received for these neoplasms, which were mostly not or not fully malignant, contributed to osteosarcoma development. This is now self-evident when assessing text and tables. We have therefore decided to not amend the text any further.
Reviewer 2 Report
Comments and Suggestions for Authors
In this manuscript entitled “Osteosarcoma arising as a secondary malignancy following treatment for a hematologic cancer. A report of 33 affected patients from the Cooperative Osteosarcoma Study Group (COSS).”, Bielak et al. focused on osteosarcomas occurring secondarily to hematologic malignancies treated with radiation therapy. This is a novel study that demonstrated a higher risk of development of secondary osteosarcomas in pediatric patients who have undergone radiation therapy for leukemia or lymphoma. This research is very interesting & well written.
I have no comments.
Author Response
We would like to thank all three reviewers for their efforts! We have tried to address their comments and suggestions as follows:
REVIEWER 2
Comment
“In this manuscript entitled “Osteosarcoma arising as a secondary malignancy following treatment for a hematologic cancer. A report of 33 affected patients from the Cooperative Osteosarcoma Study Group (COSS).”, Bielak et al. focused on osteosarcomas occurring secondarily to hematologic malignancies treated with radiation therapy. This is a novel study that demonstrated a higher risk of development of secondary osteosarcomas in pediatric patients who have undergone radiation therapy for leukemia or lymphoma. This research is very interesting & well written.
I have no comments.”
Answer
We would like to thank reviewer 2 for this very positive reception of our work!
Reviewer 3 Report
Comments and Suggestions for Authors
Authors searched the Cooperative Osteosarcoma Study Group’s database for individuals who developed osteosarcoma following a previous hematological malignancy. 33 patients were identified (median 12.9 years at diagnosis of hematological cancer; 20 lymphomas, 13 leukemias). A cancer-predisposition syndrome was evident in 1 patient only. The hematological 40 cancers had been treated by radiotherapy in 28 and chemotherapy in 26 cases. The secondary bone sarcomas arose after a median lag-time of 9.4 years when patients were a median of 19.1 years old. Tumors were considered radiation-related in 26 cases. Osteosarcoma-sites were in the extremities (19), trunk (12), or head & neck (2). Local therapy was by surgery in 27 patients with a good response reported for 9/18 eligible patients. Local radiotherapy was given to 3 patients. Median follow-up was 3.9 years after bone-tumor diagnosis. During this period, 21 patients had developed events as defined and 15 had died, for 5-year event-free and overall-survival rates of 40% and 56% respectively. Several factors were found of prognostic value (p<.05) for event-free (osteosarcoma site in the extremities) or overall (achievement of a surgical osteosarcoma-remission, receiving chemotherapy for the hematologic malignancy) survival. The authors were able to prove radiation-therapy for hematological malignancies to be the predominant risk-factor for later osteosarcomas. An overrepresentation of axial as well as a tendency towards additional neoplasms affects prognosis.
Interesting well-written paper, prooving that radiotherapy is the main responsible for secondary osterosarcoma and needs to be avoided as most azs possible in pediatric hematologic malignancies.
Bibliography is complete.
We would be interested to know the ratio of secondary OS among primary OS in the same period.
Author Response
We would like to thank all three reviewers for their efforts! We have tried to address their comments and suggestions as follows:
REVIEWER 3
Comment 1
Authors searched the Cooperative Osteosarcoma Study Group’s database for individuals who developed osteosarcoma following a previous hematological malignancy. 33 patients were identified (median 12.9 years at diagnosis of hematological cancer; 20 lymphomas, 13 leukemias). A cancer-predisposition syndrome was evident in 1 patient only. The hematological 40 cancers had been treated by radiotherapy in 28 and chemotherapy in 26 cases. The secondary bone sarcomas arose after a median lag-time of 9.4 years when patients were a median of 19.1 years old. Tumors were considered radiation-related in 26 cases. Osteosarcoma-sites were in the extremities (19), trunk (12), or head & neck (2). Local therapy was by surgery in 27 patients with a good response reported for 9/18 eligible patients. Local radiotherapy was given to 3 patients. Median follow-up was 3.9 years after bone-tumor diagnosis. During this period, 21 patients had developed events as defined and 15 had died, for 5-year event-free and overall-survival rates of 40% and 56% respectively. Several factors were found of prognostic value (p<.05) for event-free (osteosarcoma site in the extremities) or overall (achievement of a surgical osteosarcoma-remission, receiving chemotherapy for the hematologic malignancy) survival. The authors were able to prove radiation-therapy for hematological malignancies to be the predominant risk-factor for later osteosarcomas. An overrepresentation of axial as well as a tendency towards additional neoplasms affects prognosis.
Interesting well-written paper, prooving that radiotherapy is the main responsible for secondary osterosarcoma and needs to be avoided as most azs possible in pediatric hematologic malignancies.
Bibliography is complete.
Answer 1
That is a correct summary and we would like to thank the reviewer for the very positive reception of these findings!
Comment 2
We would be interested to know the ratio of secondary OS among primary OS in the same period.
Answer 2
We are very sorry but the database used for this analysis was limited to patients experiencing a leukemia or lymphoma as a secondary event. Hence, we cannot make any statements regarding patients who did not experience such secondaries or provide details about the relative epidemiology.
More information on the incidence of secondary malignant neoplasms after sarcoma in general might be found in a recent paper by Kube et al. (Cancer 2022, 128:1787-1800). In this paper evaluating the incidence of secondary malignant diseases in all German pediatric sarcoma studies until 2012, the incidence in osteosarcomas was reported as 1.8% (40/2.166). However, this report gave no data on the specific incidence of secondary hematologic neoplasms in former osteosarcoma patients. After quite some deliberation, we have therefore decided that this paper does not add specific information valuable enough to be included and referenced in the present report. We hope the reviewer can agree with this decision. If not, we would of course be willing to try and come up with some more or less meaningful statement regarding this paper and this issue.
Round 2
Reviewer 1 Report
Comments and Suggestions for Authors
Thank you for revising the manuscript and for your responses. I have no additional suggestions.